

# Significant multi-decadal variability of German wind energy generation

Jan Wohland[1,2], Nour Eddine Omrani[3], Noel Keenlyside[3], and Dirk Witthaut[1,2]

[1]Forschungszentrum Jülich, Institute for Energy and Climate Research – Systems Analysis and Technology Evaluation, 52428 Jülich, Germany
[2]University of Cologne, Institute for Theoretical Physics, 50937 Cologne, Germany
[3]University of Bergen, Geophysical Institute and Bjerknes Centre for Climate Research, Bergen, Norway

**Correspondence:** Jan Wohland (j.wohland@fz-juelich.de)

**Abstract.** Wind energy has seen large deployment and substantial cost reductions over the last decades. Further ambitious upscaling is urgently needed to keep the goals of the Paris Agreement within reach. While the variability of wind power generation poses a challenge to grid integration, much progress in quantifying, understanding and managing it has been made over the last years. Despite this progress, relevant modes of variability in energy generation have been overlooked. Based on

long-term reanalyses of the 20th century, we demonstrate that multi-decadal wind variability has significant impact on wind energy generation in Germany. These modes of variability can not be detected in modern reanalyses that are typically used for energy applications due to their short covered timespan of around 40 years. We show that energy generation over a 20y wind park lifetime varies by around ± 5% and the summer-to-winter ratio varies by around ± 15%. Moreover, ERA-interim based annual and winter generations are biased high as the period 1979 - 2010 overlaps with a multi-decadal maximum of wind

energy generation. The induced variations of windpark lifetime revenues are at the order of 10% with direct implications for profitability. Our results suggest to rethink energy system design as a perpetual process. Revenues and seasonalities change on a multi-decadal timescale, and so does the optimum energy system layout.

## 1 Introduction

Wind energy is on the rise. Following a period of high subsidies, drops in wind energy costs have been dramatic. In some places, onshore wind energy outperforms all other types of power generation in terms of levelized costs of electricity (IEA and IRENA, 2017). This economic development, in conjunction with the necessity to eliminate carbon emissions from the electricity sector in the next decades (Schleussner et al., 2016; Rogelj et al., 2015), will most certainly lead to strong investments in wind energy.

Wind parks are costly long-term investments. Since 2000, almost b€ 95 have been invested in wind parks in Germany
(BMWi, 2018). Compared to current stock exchange values, this figure is higher than the value of Volkswagen and only



marginally lower than that of Germany's most valuable company SAP (PWC, 2018). While planning is typically based on 20 year lifetimes, real-world experiences suggest that turbines can be operated even longer (Ziegler et al., 2018).

Wind power generation is variable which complicates its integration into power systems. This fact is increasingly accounted for in energy system models (a recent overview is provided by Ringkjøb et al., 2018). Underlying wind generation timeseries
are typically based on modern reanalysis (e.g., Gonzalez Aparcio et al., 2016; Staffell and Pfenninger, 2016; Moraes et al., 2018). These timeseries cover around 40 years as the observations that they rely on become available in the late 1970s. Many characteristics of renewable generation variability, such as monthly, seasonal and even decadal variability can be investigated using these datasets. But are 40 years sufficient to capture all relevant modes of wind variability?

Some components of the climate system vary on very long timescales and interactions can give rise to low-frequency vari-
ability of atmospheric processes. For example, the North Atlantic Oscillation (NAO) has a low-frequency component that is linked to ocean and stratospheric variability (Omrani et al., 2016). The NAO has also been shown to impact the British wind sector (Brayshaw et al., 2011; Ely et al., 2013) and solar generation in Iberia (Jerez et al., 2013). These links suggest that renewable power systems could be affected by low-frequency climate variability. While much attention has been given to the impacts of climate change on renewable power systems (e.g., Pryor and Barthelmie, 2010; Reyers et al., 2016; Tobin et al.,
2016; Wohland et al., 2017; Weber et al., 2018; Schlott et al., 2018; Karnauskas et al., 2018; Jerez et al., 2019), little emphasis has been put on the natural low-frequency variability of wind energy (with the notable exception of  Bett et al., 2013, 2017). The fact that climate change assessments unanimously report relatively small to negligible impacts of climate change in Europe does not necessarily imply that natural variability is insignificant because climate models exhibit major discrepancies in simulating low-frequency climate variability (e.g., Ba et al., 2014).
In this study, we investigate the long-term evolution of wind energy generation in Germany. We aim to verify if there are relevant modes of variability on timescales of multiple decades. If these modes exist, it is crucially important to incorporate them in long-term decision making with regard to the design and operation of future power systems. Moreover, they would not only matter on a system level but also affect individual investments.

## 2   Methods and data

Our focus is on the effect of long-term natural climate variability on wind power generation. To isolate the imprint of the climate, we neglect potential changes in technology and deployment of wind parks. Specifically, we freeze the current configuration of wind parks and compute their theoretical energy generation over the 20th century. This approach allows to quantify the importance of climate driven multidecadal variability of wind energy in Germany.

We derive nationally aggregated wind generation timeseries for the period 1901-2010 following the procedure detailed in
Wohland et al. (2018a). In short, the method consists of: vertical extrapolation of 10m wind speeds to 80m hub height using a power law followed by the application of a standard turbine power curve at each grid point and finally a multiplication with the installed capacities (from OPSD, 2017). Projections of the installed capacities onto the grids of the 20th century reanalyses are shown in Fig. 1.




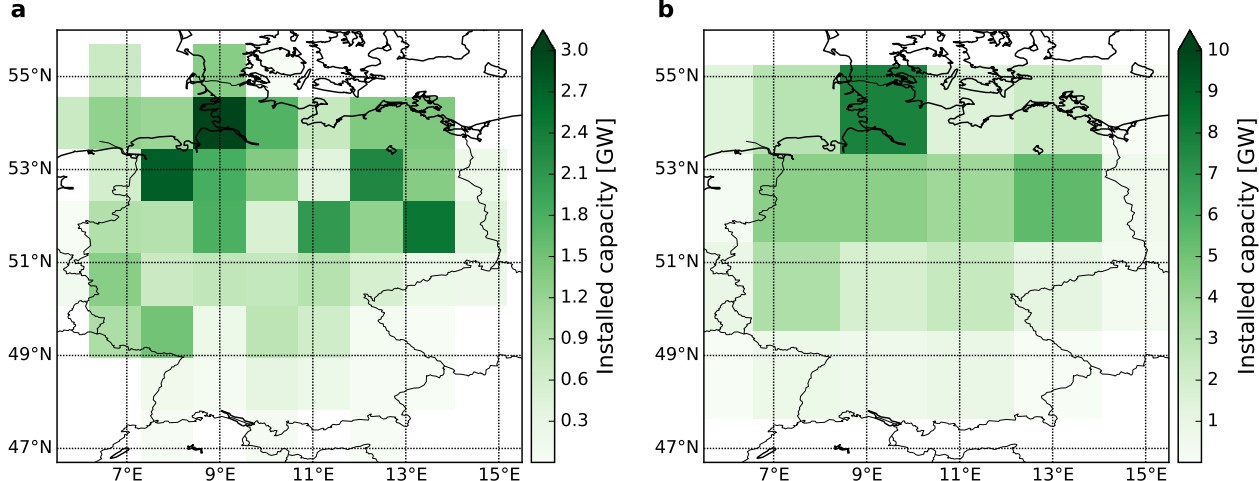

**Figure 1. Allocation of turbines based on the Open Power System Database for the end of 2016 (OPSD, 2017).** Data is projected on the ERA20CM/ERA20C/CERA20C grid (**a**) and the 20CR grid (**b**).

## 2.1  20th century reanalyses

Wind speeds come from the full set of current 20th century reanalyses and are provided by two different centers: the European Centre for Medium-Range Weather Forecast (ECMWF) and the National Oceanic and Atmospheric Administration (NOAA) from the USA. NOAA provided the first 20th century reanalysis named 20CR (Compo et al., 2011). 20CR is an atmospheric

reanalysis that assimilates sea-level pressure observations only. In this study, we use the ensemble mean wind speeds from the version 20CRv2c which has 58 ensemble members. ECMWF followed a different approach and assimilates both sea-level pressure and marine wind observations. This difference in approaches yields substantial disagreement with respect to long-term wind speed trends (Wohland et al., 2019) but, as we show, there is large agreement regarding seasonal to multi-decadal variability after subtraction of the linear trends. ECMWF provides an atmosphere (ERA20C, Poli et al., 2016) and a coupled

atmosphere-ocean 20th century reanalysis (CERA20C, Laloyaux et al., 2018). ERA20C is deterministic (i.e., has only one member) and CERA20C comes with a ten member ensemble. Unless otherwise stated, we report the CERA20C ensemble mean as the spread is usually very limited.

The longer temporal coverage comes at the cost of reduced spatial resolution as compared with modern reanalyses such as ERAINT (Dee et al., 2011), MERRA/MERRA2 (Rienecker et al., 2011) or ERA5 (Hennermann, 2018). ERA20C and

CERA20C have a spatial resolution of 1.125° x 1.125° and the 20CR resolution is even coarser (1.875° x 1.875°). While the datasets are thus clearly not well suited for site-specific assessments, they are sufficiently detailed for country-level assessments (see also Fig. 1). Temporal resolution is 3h for all datasets and hence allows to capture intra-day effects.





## 2.2 Trend removal and timescale of interest

There is demonstrated disagreement of the 20th century reanalyses in terms of wind speed trends which originates from the assmimilation of marine winds by ECMWF (Wohland et al., 2019). We thus remove the long-term (1901 - 2010) trends by subtraction of the zero-mean trend that is obtained via least-squares fitting of a linear fit function and subsequent subtraction
of the trends mean:

$$G(t) = G_{\text{raw}}(t) - (G_{\text{trend}}(t) - \langle G_{\text{trend}}(t) \rangle), \tag{1}$$

where $G_{\text{raw}}(t)$ denotes the raw annual or seasonal timeseries, $G_{\text{trend}}(t)$ denotes the linear fit and $\langle G_{\text{trend}}(t) \rangle$ is its mean value.

We focus on the long term evolution of 20 year generation averages because 20 years are a typical lifetime for wind parks. Moreover, the averaging smooths the pronounced interannual variability which has already been extensively studied elsewhere.
Both energy system planning and wind park investment are forward procedures in the sense that infrastructure built today will be operated under the weather conditions of the future. We therefore decided to compute 20 year forward running means of wind power generation $G_{20}$ as

$$G_{20}(t) = \frac{1}{20} \sum_{t'=t}^{t+20y} G(t'), \tag{2}$$

where $G(t')$ denotes the annual wind power generation in year $t'$. To study the evolution in different seasons (winter DJF,
spring MAM, summer JJA, autumn SON), we similarly compute the seasonal 20 year means as

$$G_{20}^{\text{season}}(t) = \frac{1}{20} \sum_{t'=t}^{t+20y} G(t')^{\text{season}}, \tag{3}$$

where $G(t')^{\text{season}}$ denotes the wind power generation in the respective season of year $t'$. Note that $G_{20}(t)$ and $G_{20}^{\text{season}}(t)$ are ill defined at the end of the dataset when 20 years are not available. We thus only compute them up to 1990. We generally report normalized lifetime generation or normalized seasonal lifetime generation which is obtained by division of $G_{20}(t)$ or
$G_{20}^{\text{season}}(t)$ with the 1901-2010 mean $\langle G(t) \rangle$ or $\langle G(t) \rangle^{\text{season}}$, respectively.

### 2.2.1 Seasonality

In addition to seasonal generation averages, we report the seasonality $S$, which we define as the ratio of normalized winter to summer generation:

$$S(t) = \frac{G_{20}^{\text{DJF}}(t)}{\langle G \rangle^{\text{DJF}}} \bigg/ \frac{G_{20}^{\text{JJA}}(t)}{\langle G \rangle^{\text{JJA}}}. \tag{4}$$

Seasonality is an important metric for power system design and has a large influence on optimum technology mixes (e.g., Heide et al., 2010). In Germany, wind power generation is generally higher in autumn and winter than in spring and summer.



To ensure stable operation of the power system (i.e., a balance of generation and demand at all timesteps), seasonality has to be accounted for in power system design. For example, the dimensioning of storage or backup infrastructure and optimum wind to solar mixes depend on the seasonality. For completeness, we provide an extended definition of seasonality $\hat{S}$, which also includes autumn and spring as

$$\hat{S}(t) = \frac{G_{20}^{\text{SON+DJF}}(t)}{\langle G \rangle^{\text{SON+DJF}}} \bigg/ \frac{G_{20}^{\text{MAM+JJA}}(t)}{\langle G \rangle^{\text{MAM+JJA}}}. \tag{5}$$

### 2.2.2 Bias

We use the term bias to assess whether the period covered by ERAINT is representative for the longer period covered by the 20th century reanalyses. For example, if the seasonality over 1979 – 2010 is higher than over 1900 – 2010, we call the seasonality estimates of modern reanalyses biased high.

### 2.3 Multi-taper spectral estimation

We test significance of low-frequency components in the annual and seasonal wind generation timeseries using the multi-taper method (MTM, Ghil, 2002). Classical approaches, such as Fourier spectral analysis suffer from spectral leakage when applied to relatively short timeseries, hindering reliable assessments. MTM provides an alternative in that it calculates tapers that are designed to minimize leakage. We use $K = 3$ tapers with a bandwidth of $p = 2$y as suggested by Ghil (2002) for a comparable timeseries. Eigentapers are weighted based on their eigenvalues and the computation is performed via the Python package spectrum (Cokelaer and Hasch, 2017)

Significance testing is based on the null hypothesis of red noise. The underlying process that creates a red-noise spectrum is referred to as a autoregressive model of first order or AR(1). The parameters of the red-noise spectrum $S_R(f)$ are fitted to minimize the mismatch between the median smoothed real and the red-noise spectrum (as suggested by Ghil, 2002; Mann and Lees, 1996). A peak in the real spectrum $S(f)$ at frequency $f'$ is considered significant at the 90% level if

$$S(f') > S_R(f') \cdot \chi^2(90\%, 2K), \tag{6}$$

again following (Ghil, 2002). $\chi^2(90\%, 2K)$ denotes the chi square distribution with $2K$ degrees of freedom at a 90% confidence level. White noise is a special case of red noise and is characterized by a constant spectrum (i.e., $S_W(f) = S_0$, where $S_0$ is a real positive number). White noise is generated by an autoregressive model of 0th order, AR(0).

### 2.4 Impacts on investments

In an investment decision, the installation and operational costs of an asset have to be compared with expected revenues. Taking into account risks and alternative investments, an investment is made if the expected revenues exceed the total costs by some amount. The expected revenue may be substantially flawed if it is based on only a couple of years of wind data. In contrast, decision makers that are aware of all modes of wind variability gain an advantage through more reliable revenue estimates.



To quantify this impact of low-frequency wind variability on wind park investments, we calculate the discounted lifetime cash inflows as

$$C_{\text{in}}(t) = c \cdot \sum_{t'=t}^{t+\tau} \frac{1}{(1+\gamma+\Delta\eta)^{t'-t}} G(t'),\tag{7}$$

where $\gamma = 5.5\%/y$ is the discount rate, $\Delta\eta \approx 1.5\%\%/y$ accounts for the decline in turbine performance (Staffell and Green, 2014), $\tau = 20y$ is the conservatively assumed lifetime, $c$ is the revenue per generated unit of electricity and $G$ is wind power generation. We set $c$ to be constant because the German system is still designed to guarantee prices for wind park operators. Prior to the recent shift towards auctions, the price was determined politically. Since the latest reform of the renewable energy act in 2017, the price is determined via auctions but is still guaranteed over 20 years (BMWi, 2017). Both for old and new wind parks it is thus justified to use constant prices, albeit the price will differ dependent on the date of construction and the auction outcome.

## 2.5 North Atlantic Oscillation

To gain more insight into the co-evolution of wind generation variability and the general circulation of the atmosphere, we include the North Atlantic Oscillation (NAO). The NAO is the leading pattern of climate variability in the North Atlantic Sector affecting weather and climate over Europe, particularly in winter (Marshall et al., 2001). It is here defined as the first principle component of sea-level pressure over the area 20°N to 80°N and 90°W to 40°E as detailed in Omrani et al. (2016). Our NAO index is computed from sea-level pressure data from the Hadley Center (Rayner, 2003) over the winter months December, January, February.

## 3 Validation

In a recent study, we have shown that ERAINT has skill to reproduce reported wind power generation in Germany (Wohland et al., 2018a). It thus appears logical to test the 20th century reanalyses by comparison with ERAINT over the overlapping period (1979-2009). We also add the widely used Renewables.Ninja wind energy dataset that is based on MERRA-2 (Staffell and Pfenninger, 2016).

The evolution of the normalized lifetime mean generation is similar for all reanalyses under consideration (see Fig. 2a). All start with a period of high values that is followed by roughly five years of low values. Towards the end, the normalized lifetime generation recovers, but not to the same levels as in the first couple of years.

On a finer temporal scale, there is good correlation between the daily generations based on ERAINT and 20CR, ERA20C and CERA20C, respectively (see Fig. 2b–d). 20CR overestimates daily generation (slope $< 1$ in Fig. 2b). In contrast, ERA20C and CERA20C underestimate daily generation (slopes $> 1$ in Fig. 2c&d). This systematic over/underestimation of daily wind generations, however, is of minor importance in this study because it is reduced by normalization with the long-term mean. All 20th century reanalyses agree well with ERAINT for very high daily generations larger than around 40GW. Pearson correlation



is high for 20CR (r = 0.92) and even higher for the ECMWF products (r = 0.98). A similar result is found for the RMSE which is 4.3 GW for 20CR and around 1.3 GW for ERA20C and CERA20C, again indicating larger agreement across the ECMWF reanalyses. This larger agreement could be due to more similar spatial resolutions that allow to capture the same processes in (C)ERA20C as in ERAINT. It may also reflect the common institutional origin as ERAINT and (C)ERA20C have been

5 developed at ECMWF and are based on different versions of the same model. In any case, the substantial agreement of the detrended timeseries on different timescales creates confidence in the 20th century reanalyses.

From visual inspection, there also seems to be a downward trend over the period (1979 - 1990). A trend analysis of the ERAINT data indeed reveals a significant (at the 99% level) downward trend of the normalized lifetime generation, highlighting the relevance of long-term assessments. However, this trend should be interpreted cautiously as it is calculated using only 11

10 (not independent) values of $G_{20}$. The remainder of the paper is therefore based on longer timeseries to allow more robust assessments of multi-decadal variability.





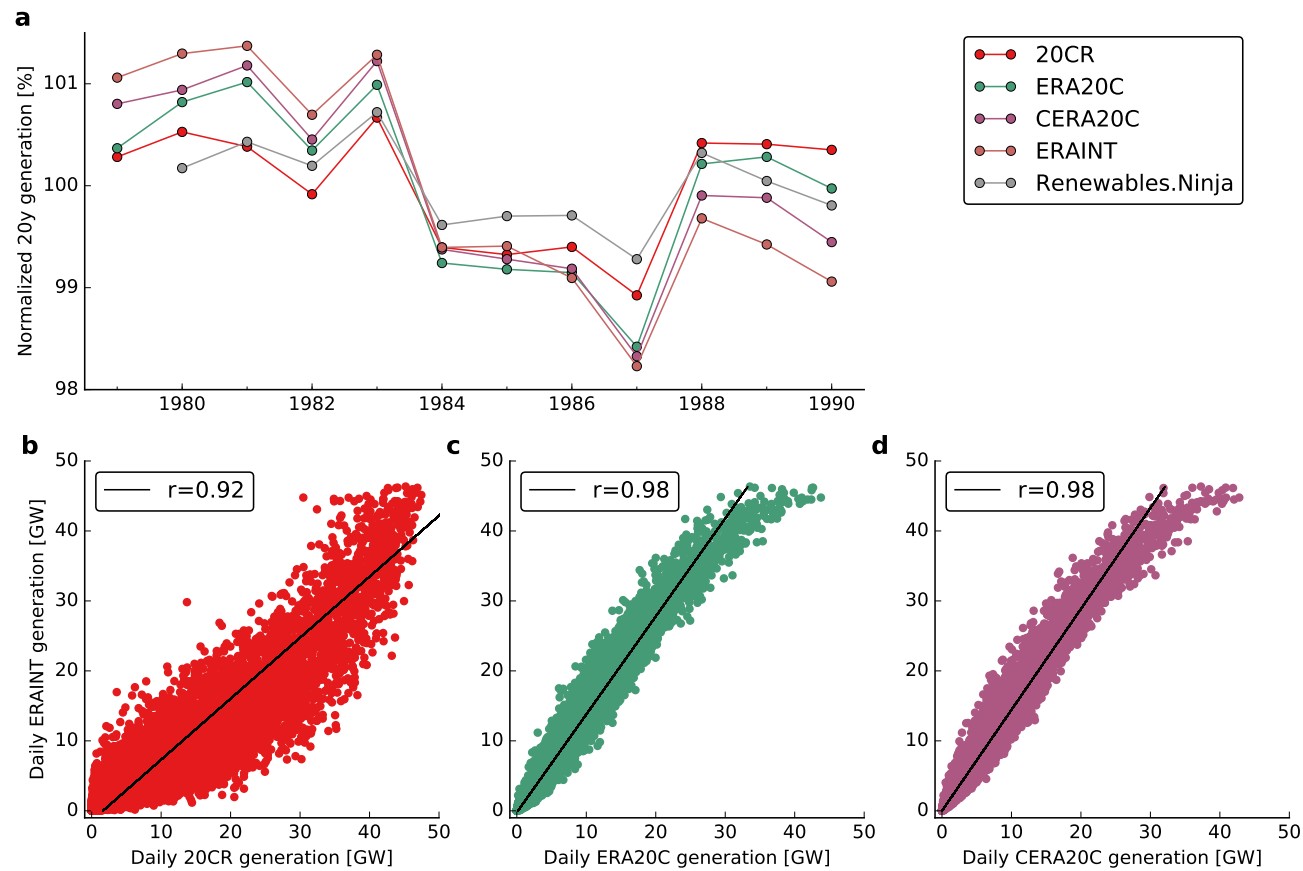

**Figure 2. German wind power generation from modern reanalyses (ERAINT, MERRA2) and 20th century reanalyses (20CR, ERA20C, CERA20C) for period of overlap. a)** Normalized lifetime generation (i.e., the reported value for 1990 is the average wind power generation of the years 1990-2009 normalized with the long-term mean). Renewables.Ninja is an openly available generation dataset that is based on MERRA2. **b-d)** scatter plots of daily generation from ERAINT versus daily values from 20CR (b), ERA20C (c), CERA20C (d) for the 30y period from 1979 to 2009. The Pearson correlation coefficient $r$ between the daily data is given in the legends. The data is shown prior to long-term trend removal which was performed for the centennial analysis (see Methods).



**Table 1. Trend characteristics.** Slopes are rounded to integer values and the CERA20C slope corresponds to the mean of the slopes of the individual ensemble members. Significance is tested against the null hypothesis of no trend and using a two-sided t-test. For CERA20C, all streams feature significant trends individually.

| dataset | slope [%/100y] | significant at 99.9% level? |
|---------|----------------|------------------------------|
| 20CR version | 0 | no |
| ERA20C | 28 | yes |
| CERA20C | 16 | yes |

## 4   Results

### 4.1   Trends

We find ERA20C and CERA20C to feature statistically highly significant trends (see Table 1). In both datasets, the trends are strong: ERA20C reports 28% higher wind power generation at the end of the 20th century as compared to its beginning. The

corresponding increase in CERA20C is substantial (16% increase in a hundred years) but roughly half as large. In contrast, there is no significant trend in 20CR.

The existence of these trends comes as no surprise given strong long-term trends in (C)ERA20C surface wind speeds over large parts of the world (Wohland et al., 2019). In our previous publication, we showed that the trends originate from the assimilated marine wind speeds that also feature very strong long-term trends. They are likely spurious and caused by the

evolving measurement technique. In addition to wind speed trends, ERA20C also features trends in marine sea level pressure gradients that are not in line with observations (Bloomfield et al., 2018). All following analyses are therefore based on detrended timeseries.

### 4.2   Low-frequency variability in normalized lifetime wind generation

After subtraction of the trends, there is large agreement among the datasets regarding multi-decadal variability of normalized

lifetime generation (see Fig. 3). Maxima and minima of annual and seasonal timeseries coincide for ERA20C, CERA20C and 20CR. The amplitude of variability is also comparable among the datasets for all seasons and the annual values. Only in September-October-November (SON), there is disagreement from 1960 onwards as 20CR reports values that are 5 to 10% off the (C)ERA20C values. Generally, there is stronger variability of seasonal than annual generation, hinting at compensating effects between seasons. In June-July-August (JJA), for example, the maximum to minimum difference is around 15%. This

compares to 5 to 10% maximum to minimum difference for the annual values.

German annual generation is dominated by winter generation due to generally stronger winds in winter. This winter dependence explains the high similarity between the annual and winter timeseries (compare Fig. 3 **a** with **c**) and also the high correlation of $r = 0.71$ between them (see Fig. 3 **b**). On the timescales considered here, there is also a weak anti-correlation between the annual and the summer values ($r = -0.39$) and between the summer and autumn values ($r = -0.46$).





The ratio of winter to summer generation (i.e., seasonality) is characterized by strong multi-decadal variability. While the maximum 20 year seasonality is between 110% and almost 120% (dependent on the dataset), the minimum lies between 80% and 90% (see Fig. 3g). If an extended definition of seasonality is applied, the amplitude of the variability is reduced but the maximum to minimum difference still ranges around 15% to 20% (see Fig. 3h).

In winter there is also a good connection between 20 year mean anomalies of the North Atlantic Oscillation (NAO) and normalized lifetime generation as highlighted by correlation coefficients between them that range from $r = 0.7$ to $r = 0.76$ for the different datasets (see Fig. 4a). This relation is consistent with the NAO being the dominant pattern of winter time climate variability in the North Atlantic sector (Marshall et al., 2001). The agreement is strongest on multidecadal timescales and it is particularly high since 1960. However, a peak in normalized lifetime wind generation around mid century is not paralleled by

a similar feature in the NAO.

    Modern reanalyses, such as ERAINT, are too short to capture these modes of low-frequency variability (see blue arrows in Fig. 3). Unfortunately, ERAINT does not only fail to capture these effects, but also provides biased high estimates in some cases. For example, the seasonality reported by ERAINT, coincides with above average values of seasonality and is hence not representative in general (see Fig. 3g). The same is true for annual and winter generation. Moreover, ERAINT begins at a time

of maximum normalized lifetime wind generation. ERAINT based trend assessments can thus misidentify the downward part of reoccurring cycles as trends (as discussed in Sec. 3). Similarly, the decline of autumn generation since the 1970s could be falsely interpreted as a trend.





**Figure 3. Normalized lifetime generation from German wind parks.** Timeseries are based on detrendeded 20th century reanalyses. The subplots show annual (**a**) and seasonal (**c-f**) timeseries. Different versions of the seasonality are also displayed (**g-h**) and correlations between seasons are reported for ERA20C (**b**). The data has been smoothed by application of a running mean 20y forward filter (i.e., the reported value for 1900 is the average of the years 1900-1919). The blue arrow highlights the limited coverage of ERAINT.





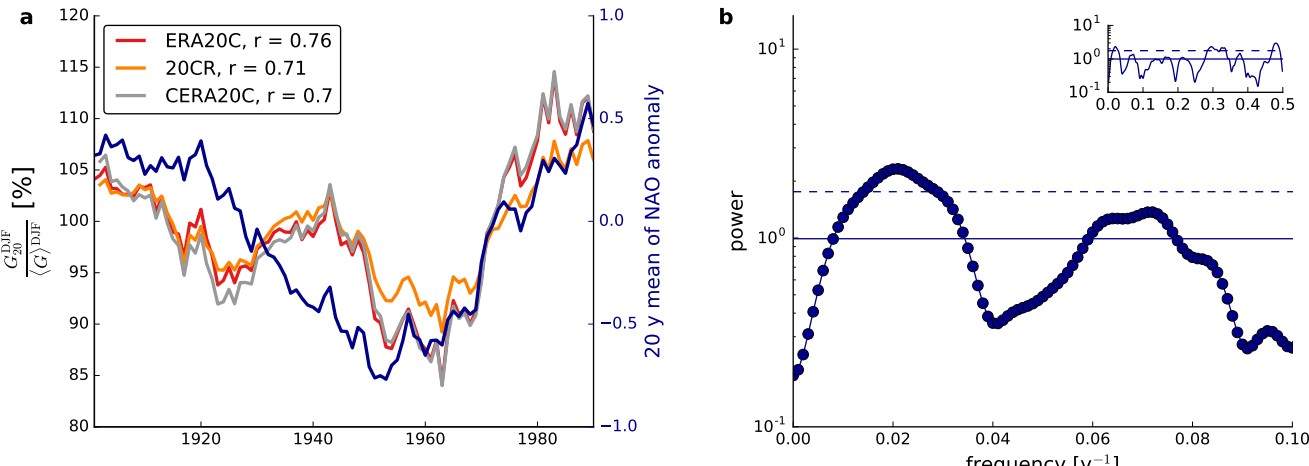

**Figure 4. Relation between normalized lifetime winter generation and the winter North Atlantic Oscillation.** Timeseries of wind power generation (in red, orange and grey) refer to the left y-axis while the NAO timeseries (in blue) refers to the right y-axis (**a**). Pearson correlation coefficients $r$ are calculated between the 20y mean NAO anomaly and the 20y mean DJF wind power generation. MTM spectrum of the winter NAO (bullets in **b**), focusing on the low-frequency interval of the spectrum. Solid lines represent the fitted spectrum of an AR(1) process that is used for significance testing and the dashed lines correspond to the 90% confidence level (see Methods for details).



### 4.3 Spectral analysis

We perform multi-taper spectral analysis for detrended annual and seasonal German wind power generation for the period 1901-2010 (Fig 5). No prior smoothing or filtering is applied. A focus is given to the low frequency part of the spectrum with frequencies of less than $0.1$ y$^{-1}$, which corresponds to at least 10 year periods. There are statistically significant low frequency

peaks in all seasons with different levels of agreement among reanalyses. All reanalyses feature a significant peak in MAM ($f \approx 0.04$y$^{-1}$ or $f^{-1} \approx 25$y) and JJA ($f \approx 0.03$y$^{-1}$ or $f^{-1} \approx 33$y) and the latter is also clearly visible in the timeseries (see Fig. 3**e**). In SON, CERA20C and ERA20C report a clearly significant peak that is also almost significant in 20CR ($f \approx 0.02$y$^{-1}$ or $f^{-1} \approx 50$y). In winter there is a spectral peak with period of around 50 years ($f \approx 0.02y^{-1}$) that is related to the NAO (see Fig.4**b**). This connection to a physical pattern of climate variability suggests that the peak is not a statistical artifact, despite

its low statistical significance. The generally high agreement among the reanalyses adds confidence to the existence of multi-decadal periodicities during the historical period.

Interestingly, the AR(1) fit to the median-smoothed spectra does not reveal red noise but white noise (except for MAM), in agreement with the understanding of atmospheric variability as a process that is white to first order (Wunsch, 1999). This can be seen by the thin solid lines in Fig. 5, which display the fitted AR(1) spectra: They are virtually flat, i.e. virtually independent

of the frequency. For example, in JJA (Fig. 5d), the power of the AR(1) fit is $10^0$ (GWh/GWh)$^2$ for all frequencies. White noise implies that the system does not have relevant memory from one year to the next but rather behaves erratically on year-to-year timescales.



**Figure 5. Spectral analysis of the wind generation timeseries using multi-taper method (MTM).** Subplots report annual (**a**) and seasonal spectra (**b-e**). Focus is given to the low-frequency component with frequencies of less than 0.1 y$^{-1}$ while the full spectrum is shown in the inset of each subplot. Solid lines represent the fitted spectrum of an AR(1) process that is used for significance testing and the dashed lines correspond to the 90% confidence level for each dataset (see Methods for details).




### 4.4 Relevance for investment decisions

In addition to the relevance of low-frequency variability for system design, the long lifetime of wind parks makes returns on individual investment susceptible to low-frequency variability and not taking this susceptibility into account has substantial economic implications. The effect is illustrated in Fig. 6, where the discounted lifetime cash inflow of a wind park that follows
5   the German mean wind generation is shown. The values are normalized such that 100% refers to the 1901-2010 mean. This graph shows variability of a wind park's cash inflows between a maximum of 104% to 107% and a minimum of 95% to 97% dependent on the phase of low-frequency climate variability at the commissioning date. In other words, a wind park created in 1955 would produce 7-12% less revenue than one created in 1975. Recall that we abstract from technology innovations throughout the entire manuscript. Dependent on the individual project characteristics, most notably the ratio of the investment
10   to the *expected* lifetime cash inflows, a few per cent more or less on the income side can turn an average project into a very profitable one or might leave a slightly profitable project unprofitable. Roughly between 1960 and 1975, there was a linear increase of cash inflows which has been followed by a decrease since 1980. Assessments based on ERAINT may tend to overestimate discounted lifetime cash inflows as ERAINT coincides with a period of high wind generation.

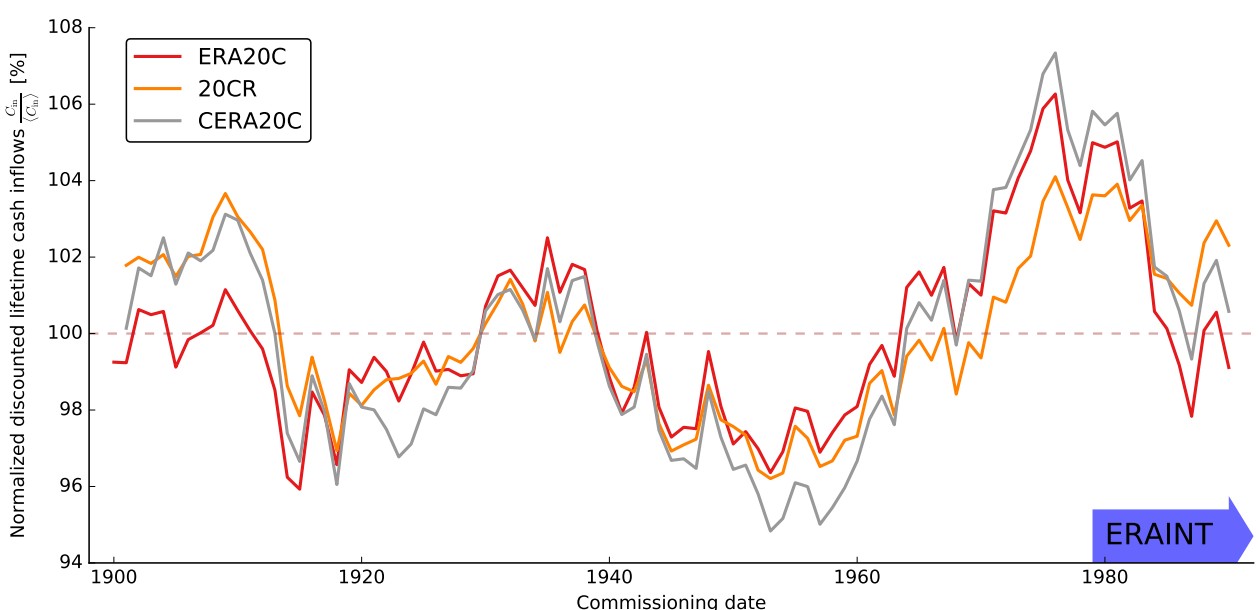

**Figure 6. Long-term evolution of normalized discounted lifetime cash inflows of a wind park whose generation follows the German mean.** A lifetime of 20y, ageing of $1.5\%/y$ and a discount rate of $5.5\%/y$ are assumed. The timeseries ends in 1990 because the underlying reanalyses end in 2010.



## 5   Discussion and concluding remarks

Based on the full set of current 20th century reanalyses (20CR, ERA20C, CERA20C), we have shown that multi-decadal variability matters for wind energy in Germany. There are statistically significant modes of generation variability on timescales of 25 to 50 years in spring, summer and autumn. In winter, there is a spectral peak with period of around 50 years that is related to the NAO. This connection to a physical pattern of climate variability suggests that the peak is not a statistical artifact, despite its low statistical significance.

Our results imply that in addition to relatively intuitive timescales (diurnal, seasonal, annual), also slower and less intuitive modes of variability ought to be included in energy assessments. While current modern reanalyses are too short to capture multi-decadal wind generation variability, future products may be better suited due to extended temporal coverage (e.g., ERA5 will start in 1950 and is expected to be entirely published in early 2019).

One of the most relevant results for power system design is the variability of seasonality (defined as the ratio of winter to summer generation here). Far from being constant, 20 year mean seasonality varies by almost up to $\pm15\%$. As the seasonal evolution of generation is one main factor to determine optimal contributions of wind and photovoltaics (Heide et al., 2010), such optimum shares should also be considered as timeseries that vary on timescales of 50 years or so. This variability calls for a perpetual redesign of power systems to follow climate variability. ERAINT samples a seasonality maximum and therefore reports biased high seasonality. This bias implies that lifetime wind power generation is most often more stable throughout the year than would be expected from ERAINT, facilitating system integration. In the bigger picture, it may be relevant to rethink whether changes in seasonality that were attributed to climate change in earlier studies (e.g., Reyers et al., 2016) may simply reflect natural variability.

There are also implications for individual wind park projects as their profitability is strongly influenced by climate variability on long timescales. The same wind park commissioned in different phases of low-frequency generation variability, can have discounted lifetime cash inflows anywhere between 95% and 107% of the mean value with potentially severe impacts on profitability. To give an impression of scale: As the current German wind park fleet represents a b€ 95 investment, this translates into a lifetime revenue spread at the order of b€ 10 in Germany alone.

The effect of wind variability on revenues obviously depends on the market design. Instead of guaranteeing a constant price for wind energy, adaptive prices that fall in times of high generation and decline in times of low generation could dampen the economic effect of multi-decadal wind variability. We speculate that a higher price of $CO_2$ emission allowances in combination with an end to guaranteed renewable feed-in might be a possible route forward. The increased $CO_2$ emission allowance price would guarantee that renewables are favoured over fossils for mere economic reasons and it would also ensure sufficiently high market prices. During decades of high (low) wind generation, the average market price would fall (increase) thereby smoothing the variability of revenues and reducing the risk for investors. However, this strategy would only constitute an interim solution as it relies on a substantial share of non-renewable generation. In a future zero emission energy system, all variability from wind generation needs to be balanced by other means, for example through sector coupling, flexible demands or large scale




storage (Brown et al., 2018). It might become necessary to ponder decadal energy storage systems or to use the atmosphere as a carbon storage (Wohland et al., 2018b).

Our study raises new questions. While Germany was chosen as an exemplary case due to its current high deployment of wind turbines, other and larger areas should also be studied. Are there compensating effects across Europe? If yes, expansion of the transmission network and optimized siting could help mitigate multi-decadal variability in the same fashion that it helps to smooth synoptic generation variability (e.g., Rodriguez et al., 2014; Grams et al., 2017; Santos-Alamillos et al., 2017). This study is restricted to wind energy because we doubt the reanalyses' skill to capture cloud dynamics sufficiently well. Nevertheless, it would be of high interest to investigate low frequency variability of other types of renewable generation: Do similar modes exist for photovoltaics and hydropower? Lastly, climate models are, in theory, an excellent tool to quantify and study natural climate variability as timeseries of arbitrary length can be obtained. Multi-decadal variability can thus be sampled substantially better than in 20th century reanalyses. However, it remains to be shown whether climate models are capable to reproduce multi-decadal variability that is relevant for the energy sector.

*Code and data availability.* This paper relies on wind speeds from different 20th century reanalysis that are openly available through ECMWF (https://apps.ecmwf.int/datasets/ and http://apps.ecmwf.int/datasets/data/era20c-daily/levtype=sfc/type=an/) and NOAA (https://www.esrl.noaa.gov/psd/data/gridded/data.20thC_ReanV2c.monolevel.html). We also use the end of 2016 wind fleet configuration as reported by the Open Power System Database which is also openly available online (https://data.open-power-system-data.org/renewable_power_plants/). The programming is done in Python and the code is shared upon reasonable request to the authors.

*Author contributions.* JW initiated the collaboration, developed the methodology, analyzed the data, produced all figures and wrote most of the manuscript. DW contributed to the methodology and interpretation of the data and supervised the research. NEO and NK contributed to the development of the methodology and the interpretation of the results. All authors contributed ideas, gave feedback and helped to improve the manuscript.

*Competing interests.* The authors declare that they have no conflict of interest.

*Acknowledgements.* We thank ECMWF for making ERA20C and CERA20C publicly available. We thank NOAA for making 20CRv2c available. JW thanks the Hitec graduate school at Forschungszentrum Jülich for a travel grant. JW and DW are funded by the Helmholtz association through the the joint initiative 'Energy System 2050 – A Contribution of the Research Field Energy' and the grant no. VH-NG-1025 to Dirk Witthaut.



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
