# Peer review of "Significant multi-decadal variability of German wind energy generation"

_Wind Energy Science, 2019_

## Referee Comment (RC1) · Anonymous Referee #1 · 8 Apr 2019

This manuscript addresses an important subject and helps in the "Global Stilling" debate. The analysis of wind trends has to separate between multi-decadal trends such as those forced by the North Atlantic Oscillation and the (expected) reduction in overall wind speeds due to the decrease of the polar-tropical temperature gradient. The reduction of wind speeds in Central Europe has implications on the economic benefits from wind turbines erected during the (necessary) transition process towards a fossil-free energy infrastructure.

Unfortunately, the manuscript is not well balanced. Title, Abstract and Introduction rise the expectation that this paper mainly deals with the separation between multi-decadal variability and Climate Change. But only half of the conclusions reflect findings belonging to this issue. The other half of the Conclusions focusses on economic aspects

of our future energy supply and speaks about transmission lines across Europe and "decadal energy storage systems". How such systems could look like is not detailed (unfortunately).

Another conclusion is that the atmosphere could be used as a carbon storage. What does this mean? Once again, no explanation is given.

The task of a major revision of this manuscript has to be to prepare the discussion in the second half of the Conclusions by additional paragraphs in the Introduction.

**Specific Review Points**

The "Global Stilling" debate which has been brought up again recently should be addressed in the Introduction and in the Conclusions. This would help to put the results of the present manuscript into the proper perspective.

Page 1, line 20: Comparing an estimated revenue from wind energy devices to the overall value of a car company is a bit strange for a scientific paper.

Page 2, line 30-32: The evaluations in this manuscript are understood to be based on (grided) reanalysis data. Therefore, the reviewer does not understand why there is the necessitiy to extrapolate wind speeds from the 10 m level to hub height. There should be data from heights much closer to hub height. And if this large extrapolation has to be made, then the power law is not helpful. The exponent of the power law depends on height above ground, surface roughness and atmospheric stability. The surface roughness-dependency makes the exponent wind direction-dependent. Especially due to the height dependency of the power law exponent, this law is only suitable for extrapolations over small height intervals. Anyhow, the extrapolation necessity and procedure has to be described in much more detail.

Section 4.2: this Section contains interesting results. According to the title of the manuscript, the reader expects more explanations about how the founded trends are connected to the North Atlantic Oscillation. Why is there an anti-correlation between
the winter trends and the summer trends leading to such strong variations in the seasonality? And why are the frequencies in Fig. 5 varying so much with the season? (major peak at 0.02 for winter and autumn, 0.03 for summer and 0.04 for spring?) If they all relate to NAO, shouldn't they have all the same frequency dependence? Much more discussion is needed on this interesting finding!

Page 16, lines 14 to 15: Is the request for a perpetual redesign of power systems realistic? It sounds well, but how to do it?

A more appropriate title could be: "Multi-decadal atmospheric variability and power system design". Then the reader would expect what is found in the Conclusions. But it then becomes debatable whether this manuscript still fits into the spectrum of this journal.

---

## Referee Comment (RC2) · Sonia Jerez (Referee) · 11 Jun 2019

I think the paper addresses a key aspect of wind energy variability by focusing on its low frequency spectrum. The manuscript reads very well, the analysis is well designed, the conclusion are well rooted on the shown results and the figures are clear. I therefore recommend its publications almost as it is. I only have a very few minor comments:

Fig 1. Caption: ERA20CM?

Fig 2. Correlation between daily records should be computed from detrended series with removed seasonality, shouldn't it?

Table 1. 20CR version –> 20CR

[Figure]

Fig 3. Maybe good to use the same y-axes for all subplots. In the context of lines 18-20 in page 9.

Fig 4. Correlations are calculated between 20y mean anomalies, which means between 20y runmean anomalies, right? Are these means computed for the 20 years ahead (Equation 2)? In both cases?

Fig 4. Panel b maybe better at Fig 5.

Page 16, line 10. ERA5 is expected to be entirely published BY THE END OF 2019.

---

## Author Comment (AC1) · 25 Jun 2019

**Response to the reviewers**

'Significant multi-decadal variability of German wind energy yields' (wes-2019-8)

Jan Wohland, Nour Eddine Omrani, Noel Keenlyside, and Dirk Witthaut

June 25, 2019

1 Dear Julie Lundquist, Sonia Jerez and anonymous reviewer,

we want to thank you for your efforts in assessing our manuscript. We are convinced that
your critical, fair and thoughtful comments helped us a lot in further improving the manuscript.
We understand that both reviewers agree on the relevance of the topic (anonymous reviewer:
"addresses an important subject", Sonia Jerez: "paper addresses a key aspect of wind energy
variability"). Both reviewers also find the paper to fit within the scope of WES by suggesting
major or minor revisions.

At the same time, the reviewers disagree on some details of the manuscript and we discuss these details below. Please note that it was not always possible to reconcile contradicting assessments. In some cases we consequently could not modify the manuscript in line with the reviewer's suggestions as this would have led to conflict with the other reviewer's assessment (and our own).

In this response we address all comments and explain how we handled them. Throughout this text, red denotes deletions from the original manuscript and green denotes additions. Citations from the reviews are given in *italics*.

**16 Reviewer 1**

This manuscript addresses an important subject and helps in the "Global Stilling" debate. The analysis of wind trends has to separate between multi-decadal trends such as those forced by the North Atlantic Oscillation and the (expected) reduction in overall wind speeds due to the decrease of the polar-tropical temperature gradient. The reduction of wind speeds in Central Europe has implications on the economic benefits from wind turbines erected during the (necessary) transition process towards a fossil-free energy infrastructure.

**23 Response by the authors**

We thank the reviewer for his/her assessment. We believe that there might be some misunderstandings and would therfore like to clarify two aspects.

First, the paper does not provide evidence for multi-decadal trends. We mention the upward trends in (C)ERA20C (e.g., in Table 1) and discuss that they are likely spurious following the argumentation in an earlier paper by us (see p.9 ll.4f). Instead of reporting long-term trends, the paper reports long-term variability that is consistent accross all current 20th century reanalyses.

 $_{30}$  Second, we are not aware of convincing evidence for the reviewers claim of an expected

31 reduction in overall wind speeds due to a reduction of the polar-tropical temperature gradient.

 $_{32}$  To our best knowledge, the claim is in disarray with modeling studies which report different signs

33 of change in different regions rather than a general decline (e.g., Tobin et al., 2016; Karnauskas

et al., 2018). With respect to 'global stilling', we refer to our reply to specific comment 1 where we discuss global stilling. Please also note the changes of the manuscript that are reported there

36 there.

**37 Comment 1**

38 Unfortunately, the manuscript is not well balanced. Title, Abstract and Introduction rise the 39 expectation that this paper mainly deals with the separation between multi-decadal variability 40 and Climate Change. But only half of the conclusions reflect findings belonging to this issue. 41 The other half of the Conclusions focusses on economic aspects of our future energy supply and 42 speaks about transmission lines across Europe and "decadal energy storage systems". How such

43 systems could look like is not detailed (unfortunately).

**44 Response by the authors**

45 We thank the reviewer for bringing up these points. We agree that the discussion focused too

 $_{\rm 46}$   $\,$  heavily on the market design and general aspects and removed one paragraph to correct for this

47 (see below). We also extend the Introduction as suggested by the reviewer (see Comment 3 for

48 a more detailed discussion and the changes).

49 With all due respect, we disagree with the reviewer's interpretation that Title, Abstract and 50 Introduction raise the expectation of separating between multi-decadal variability and climate

51 change. Instead, the paper documents the existence of multi-decadal variability of wind energy

52 generation. This is reflected in the title and abstract, which both do not mention climate change

at all. Climate change impacts on renewables are mentioned in the introduction (ll. 13-19) in and to contactualize our finding with recent efforts in the community

order to contextualize our finding with recent efforts in the community.
 We believe that reporting the economic impact of multi-decadal variability on wind park

revenues as discussed in Sec. 4.4 and Fig. 6 is a valuable extension of the analysis. It reflects
 the information needs of the different stakeholders in wind energy. We therefore decided to keep
 this part unchanged.

**59 Changes in the manuscript**

60 lifetime revenue spread at the order of b 10 in Germany alone.

The effect of wind variability on revenues obviously depends on the market design. In-61 stead of guaranteeing a constant price for wind energy, adaptive prices that fall in times 62 of high generation and decline in times of low generation could dampen the economic 63 effect of multi-decadal wind variability. We speculate that a higher price of  $CO_2$  emis-64 sion allowances in combination with an end to guaranteed renewable feed-in might be 65 a possible route forward. The increased  $CO_2$  emission allowance price would guarantee 66 that renewables are favoured over fossils for mere economic reasons and it would also 67 ensure sufficiently high market prices. During decades of high (low) wind generation, the 68 average market price would fall (increase) thereby smoothing the variability of revenues 69 and reducing the risk for investors. However, this strategy would only constitute an in-70 terim solution as it relies on a substantial share of non-renewable generation. In a future 71 zero emission energy system, all variability from wind generation needs to be balanced by 72 other means, for example through sector coupling, flexible demands or large scale storage 73 (Brown et al., 2018). It might become necessary to ponder decadal energy storage systems 74 or to use the atmosphere as a carbon storage (Wohland et al., 2018b). 75

76 Our study raises new questions.

**77 Comment 2**

Another conclusion is that the atmosphere could be used as a carbon storage. What does this
 mean? Once again, no explanation is given.

**Response by the authors**

We thank the reviewer for this comment. We agree that this paragraph was too loosely coupled to the rest of manuscript and decided to remove it in order to put more focus on the main message of the manuscript (see Comment 1).

84 Nevertheless, we would disagree that no explanation was given. The sentence that the reviewer 85 refers to reads: "It might become necessary to ponder decadal energy storage systems or to use 86 the atmosphere as a carbon storage (Wohland et al., 2018b)". It does contain a reference 87 to another publication in which we discuss the potential co-benefits of a negative emission 88 technology called Direct Air Capture and renewables. Applied to multi-decadal generation 89 variability, Direct Air Capture could be used to remove  $CO_2$  in windy decades which would 90 allow for carbon emissions from backup power plants in calm decades.

**91 Changes in the manuscript**

92 See Comment 1.

**93 Comment 3**

The task of a major revision of this manuscript has to be to prepare the discussion in the second half of the Conclusions by additional paragraphs in the Introduction.

**96 Response by the authors**

97 We thank the reviewer for this suggestion. Following his/her comment, we extended the In-98 troduction which now contains additional information about the market design and renewable 99 portfolios. We also mention the global stilling phenomenon now (see also next comment). We 100 are confident that these changes improve the legibility of the manuscipt.

**101 Changes in the manuscript**

102 p. 2, l. 1f.

While planning is typically based on 20 year lifetimes, real-world experiences suggest that 103 turbines can be operated even longer (Ziegler et al., 2018). The current german market 104 design privileges renewables over conventional generators via a guaranteed feed-in and 105 wind park operators are compensated for congestion-related curtailment. This implies 106 that there is no market incentive for planners to increase the system-friendliness of their 107 wind parks. In particular in cases where a trade-off has to be made between total energy 108 generation and system-friendliness, planners and investors will likely prefer the former 109 over the latter. 110

111 p. 2, l. 3f.

This fact is increasingly accounted for in energy system models (a recent overview is provided by Ringkjb et al., 2018). Portfolios of different renewables and large-scale transmission can mitigate generation variability (e.g., Heide et al., 2011; Schlachtberger et al., 2017). Underlying wind ...

116 p.2, l. 15 f.

little emphasis has been put on the natural low-frequency variability of wind energy
(with the notable exception of Bett et al., 2013, 2017). Natural low-frequency variability
could also help to explain trends in surface wind speeds computed over a few decades
(commonly referred to as global stilling, Vautard et al., 2010) if the period featuring the
trend coincides with the downward sloping fraction of multi-decadal variability. The fact
that...

**123 Specific comment 1**

The "Global Stilling" debate which has been brought up again recently should be addressed in the Introduction and in the Conclusions. This would help to put the results of the present manuscript into the proper perspective.

**127 **Response by the authors**

We would like to thank the reviewer for this very thoughtful comment. As already mentioned in Comment 3, we added the global stilling phenomenon to the Introduction. We also add a short discussion in the Conclusion, as suggested.

**131 Changes in the manuscript**

132 p. 16, l. 6f.

This connection to a physical pattern of climate variability suggests that the peak is not a statistical artifact, despite its low statistical significance.

Wind power generation reached a multi-decadal maximum around 1980 implying that trend assessments starting in 1980 suffer from a sampling bias. The downward sloping fraction of multi-decadal variability should not be confused with a long-term trend and an extrapolation of the trend into the future is misleading. These results are relevant in contextualizing the global stilling phenomenon (Vautard et al., 2010). Our results imply that in

**141 Specific comment 2**

Page 1, line 20: Comparing an estimated revenue from wind energy devices to the overall value
of a car company is a bit strange for a scientific paper.

**Response by the authors**

We intended to provide an intuitive comparison for wind energy investments. We think that
such a comparison is helpful because most readers will likely find it hard to put b€95 into
perspective. In the respective sentence, we therefore compare the investments in wind energy
to the current stock market values of two major companies in Germany.

**149 Specific comment 3**

Page 2, line 30-32: The evaluations in this manuscript are understood to be based on (grided) reanalysis data. Therefore, the reviewer does not understand why there is the necessitiy to extrapolate wind speeds from the 10 m level to hub height. There should be data from heights much closer to hub height. And if this large extrapolation has to be made, then the power law is not helpful. The exponent of the power law depends on height above ground, surface roughness and atmospheric stability. The surface roughness-dependency makes the exponent wind direction-dependent. Especially due to the height dependency of the power law exponent,
this law is only suitable for extrapolations over small height intervals. Anyhow, the extrapolation
necessity and procedure has to be described in much more detail.

**Response by the authors**

We thank the reviewer for these methodological suggestions. We are fully aware of the assumption tions behind a power law with a fixed coefficient and the deficiencies of such an approach.

162 Unlike the reviewer's expectations, however, wind speeds are not available at another height 163 accross the entire ensemble. While the ECMWF reanalyses contain 100m wind speeds, 20CR 164 does not. In order to ensure comparability, we decided to apply the same methodology to all 165 datasets and therefore based the assessment on 10m winds. We add a sentence to the manuscript 166 to highlight this aspect of the approach (see below).

More importantly, we would like to emphasize that we validate our approach in Section 3 (validation) by comparison with an ERA-interim based timeseries that was validated with real generation data in 2015 and 2016. Owing to the good agreement between the 20th century reanalyses and ERA-interim, we are convinced that our approach is well suited for a country level assessment on long timescales (see also Fig. 2). In other words, the error introduced via the extrapolation from 10m to 80m does not matter here.

The methodology is detailed in another paper that we explicitly refer to in lines 29f. ("We derive nationally aggregated wind generation timeseries for the period 1901-2010 following the procedure detailed in Wohland et al. (2018a)". For the sake of brevity, we considered it more appropriate to only provide a brief overview of the procedure as the interested reader can access the details in the referenced publication.

**178 Changes in the manuscript**

179 p. 3, l. 12f.

as the spread is usually very limited. Our analysis is based on 10m wind speeds. In contrast to higher level wind speeds, they are available for all 20th century reanalyses allowing
us to apply the same methodology to all datasets and thereby ensuring comparability. We
validate the approach in Sec. 3.

184 The longer

**185 Specific Comment 4**

Section 4.2: this Section contains interesting results. According to the title of the manuscript, the reader expects more explanations about how the founded trends are connected to the North Atlantic Oscillation. Why is there an anti-correlation between the winter trends and the summer trends leading to such strong variations in the seasonality? And why are the frequencies in Fig. 5 varying so much with the season? (major peak at 0.02 for winter and autumn, 0.03 for summer and 0.04 for spring?) If they all relate to NAO, shouldnt they have all the same frequency dependence? Much more discussion is needed on this interesting finding!

**Response by the authors**

We thank the reviewer for this comment and we want to reassure the reviewer that we are very interested in developing a process understanding which might help to answer his/her questions. We believe that he/she refers to increasing or decreasing intervals of multi-decadal variability when using the term "founded trends". If this is not correct, we would like to ask the reviewer for a clarification on this point. We will not use the term "trend" here for a fraction of multidecadal variability as we find the wording potentially misleading. We do not really understand why the title suggests a link between multi-decadal variability and the NAO. The title does not mention the NAO.

So far, we do not have any process understanding why the multi-decadal variability of winter and summer generation are anti-correlated. Only in winter, we find a good connection between multi-decadal wind generation variability and multi-decadal variability of the winter NAO. Acknowledging the importance of the reviewer's comment and the overall value of process based understanding, we add two sentences in this regard to the discussion (see below).

The fact that spectral peaks occur at different frequencies in different seasons hints toward different underlying processes. We agree with the reviewer that this aspect has not been sufficiently clear yet and add a short paragraph (see below). To our best knowledge, there is no reason for all seasons to show spectral peaks at the same frequencies. We would be delighted if the reviewer could point us to any evidence or theory that points toward the neccessity of spectral peak at the same frequency.

**213 Changes in the manuscript**

214 p. 13, l. 12f.

during the historical period.

Spectral peaks generally do not exist at the same frequencies in different seasons. This implies that the relevant processes vary by season. While the winter NAO explains a large share of the winter variability, similar explanations can currently not be given for the other seasons.

Interestingly, the AR(1)

221 p. 17, l. 12f.

similar modes exist for photovoltaics and hydropower? In addition to the winter link
between wind power generation and the NAO, other connections between multi-decadal
renewable generation and large scale patterns of climate variability might exist. They
could contribute to a process based understanding and should therefore be investigated
in future work. Lastly, climate models are,

**227 Specific Comment 5**

Page 16, lines 14 to 15: Is the request for a perpetual redesign of power systems realistic? It sounds well, but how to do it?

**Response by the authors**

We thank the reviewer for this comment. With the sentence, we intended to highlight that an optimum system design does not exist due to climatic variability at timescales comparable to or even exceeding the lifetime of power system components. Instead, the optimum system itself varies with multi-decadal wind variability.

With respect to the implementation side, we clearly have no definite answers. It is our intention to underline that infrastructure decision making should use multi-decadal climatic information. Given that electricity and energy systems evolve constantly (by addittions, replacements, retirements of individual components), the energy system is in constant change which could be used to react to changes in the climatological boundary conditions.

240 We added another sentence to make this point clearer.

**241 Changes in the manuscript**

242 p. 16, l. 14f.

This variability calls for a perpetual redesign of power systems to follow climate variability. Even though the lifetime of individual power system components (e.g., transmission lines or power plants) is very long, additions, replacements and retirements occur frequently within the entire power system. These events theoretically allow for adaptive reactions to multi-decadal variability. ERAINT

**248 Specific Comment 6**

A more appropriate title could be: "Multi-decadal atmospheric variability and power system design". Then the reader would expect what is found in the Conclusions. But it then becomes debatable whether this manuscript still fits into the spectrum of this journal.

**252 Response by the authors**

We want to thank the reviewer for his/her suggestion to update the title. However, we think that his/her suggestion is less exact than the initial title as it (a) does not mention the spatial scale (Germany) and (b) does not mention the variable that is studied in the paper (wind energy generation). We would therefore prefer to stick to the initial title. We also think that the initial title does a better job at not creating wrong expectations (which the reviewer has been repeatedly commenting on).

**259 Reviewer 2 (Sonia Jerez)**

I think the paper addresses a key aspect of wind energy variability by focusing on its low frequency spectrum. The manuscript reads very well, the analysis is well designed, the conclusion are well rooted on the shown results and the figures are clear. I therefore recommend its publications almost as it is.

**264 Minor comments**

265 **1**

- 266 Fig 1. Caption: ERA20CM?
- 267 We deleted ERA20CM as it was not used in the paper.

**268 **2**

Fig 2. Correlation between daily records should be computed from detrended series with removed
 seasonality, shouldnt it?

271 Thank you for this comment, Sonia. Our answer is maybe and no.

You are correct that the correlations could be even higher if the trends were removed beforehand. However, given the fact that we only look at 30 years of overlap here, the impact of the trends is not as strong as if we would consider the entire 20th century. We therefore argue that a trend correction is not necessary here.

Second, we do not see any necessity to remove the seasonal cycle. This is because seasonal variations are synchronous in the 20th century reanalyses and in ERA-interim. A comparison between, for example, winter ERA20C and winter ERA-interim values is thus a fair comparison.

**279 **3**

280 Table 1. 20CR version  $\rightarrow$  20CR

281 Yes. Thanks.

**282 **4**

Fig 3. Maybe good to use the same y-axes for all subplots. In the context of lines 18-20 in page 9.

We agree that using the same y-axis would underline the argument that changes are larger for the seasonality and the seasonal values when compared to annual values. However, it might also create the impression that changes of the annual timeseries can be neglected. We would strongly disagree with this impression because  $\pm 5\%$  is far from negligible. For instance, the current debates about wind farm blockage shows clearly that the wind energy sector is interested in in deviations of projected and actual wind energy generation at the order of 1% (Bleeg et al., 2018).

**292 **5**

Fig 4. Correlations are calculated between 20y mean anomalies, which means between 20y
runmean anomalies, right? Are these means computed for the 20 years ahead (Equation 2)? In
both cases?

The correlations reported in Fig. 4b are calculated from the timeseries shown in Fig. 4a,c,d,e,f. The timeseries are calculated using Eq. 2 (annual values) or Eq. 3 (seasons). In both cases, a foreward running mean is computed. The normalization applied to the timeseries does not matter for the correlation as a constant scaling of the timesiers has no impact on the correlation coefficient.

**301 **6**

302 Fig 4. Panel b maybe better at Fig 5.

We believe that this is a question of style and there are good arguments for both options. Moving it to Fig. 5 would indeed mean that all spectra are shown in the same Figure. However, leaving it in Fig. 4 means that the entire argument related to the winter variability and the connection to the NAO is kept in one Figure. We prefer the second option as it emphasizes this interesting result.

308 7

309 Page 16, line 10. ERA5 is expected to be entirely published BY THE END OF 2019.

310 Thanks. We correct accordingly.

**311 References**

James Bleeg, Mark Purcell, Renzo Ruisi, and Elizabeth Traiger. Wind farm blockage and the consequences of neglecting its impact on energy production. *Energies*, 11(6):1609, 2018.

Dominik Heide, Martin Greiner, Lueder Von Bremen, and Clemens Hoffmann. Reduced
 storage and balancing needs in a fully renewable European power system with excess
 wind and solar power generation. *Renewable Energy*, 36(9):2515-2523, 2011. URL http:
 //www.sciencedirect.com/science/article/pii/S0960148111000851.

Kristopher B. Karnauskas, Julie K. Lundquist, and Lei Zhang. Southward shift of the global
wind energy resource under high carbon dioxide emissions. *Nature Geoscience*, 11(1):38–43,
January 2018. ISSN 1752-0894, 1752-0908. doi: 10.1038/s41561-017-0029-9. URL http:
//www.nature.com/articles/s41561-017-0029-9.

D.P. Schlachtberger, T. Brown, S. Schramm, and M. Greiner. The benefits of cooperation in a highly renewable European electricity network. *Energy*, 134:469-481, September 2017.
ISSN 03605442. doi: 10.1016/j.energy.2017.06.004. URL http://linkinghub.elsevier. com/retrieve/pii/S0360544217309969.

Isabelle Tobin, Sonia Jerez, Robert Vautard, Francise Thais, Erik van Meijgaard, Andreas
Prein, Michel Deque, Sven Kotlarski, Cathrine Fox Maule, Grigory Nikulin, Thomas Noel,
and Claas Teichmann. Climate change impacts on the power generation potential of a European mid-century wind farms scenario. *Environmental Research Letters*, 11(3):034013, March
2016. ISSN 1748-9326. doi: 10.1088/1748-9326/11/3/034013. URL http://stacks.iop.
org/1748-9326/11/i=3/a=034013?key=crossref.7da88a5a7c6dea354ce58294e3c7482b.

Robert Vautard, Julien Cattiaux, Pascal Yiou, Jean-Nol Thpaut, and Philippe Ciais. Northern Hemisphere atmospheric stilling partly attributed to an increase in surface roughness. Nature Geoscience, 3(11):756-761, November 2010. ISSN 1752-0894, 1752-0908. doi:
10.1038/ngeo979. URL http://www.nature.com/doifinder/10.1038/ngeo979.

---

## Editor Decision (ED1)

[revised manuscript text omitted]
_{20}^{\mathrm{DJF}}(t)}{\langle G \rangle^{\mathrm{DJF}}} \Big/ \frac{G_{20}^{\mathrm{JJA}}(t)}{\langle G \rangle^{\mathrm{JJA}}}. \tag{4}$$

5   Seasonality is an important metric for power system design and has a large influence on optimum technology mixes (e.g., Heide et al., 2010). In Germany, wind power generation is generally higher in autumn and winter than in spring and summer. To ensure stable operation of the power system (i.e., a balance of generation and demand at all timesteps), seasonality has to be accounted for in power system design. For example, the dimensioning of storage or backup infrastructure and optimum wind to solar mixes depend on the seasonality. For completeness, we provide an extended definition of seasonality $\hat{S}$, which also
10   includes autumn and spring as

$$\hat{S(t)} = \frac{G_{20}^{\mathrm{SON+DJF}}(t)}{\langle G \rangle^{\mathrm{SON+DJF}}} \Big/ \frac{G_{20}^{\mathrm{MAM+JJA}}(t)}{\langle G \rangle^{\mathrm{
[revised manuscript text omitted]